# Structural Parameter Optimization of a Tomato Robotic Harvesting Arm: Considering Collision-Free Operation Requirements

**DOI:** 10.3390/plants13223211

**Published:** 2024-11-15

**Authors:** Chuanlang Peng, Qingchun Feng, Zhengwei Guo, Yuhang Ma, Yajun Li, Yifan Zhang, Liangzheng Gao

**Affiliations:** 1College of Mechanical and Electrical Engineering, Xinjiang Agricultural University, Urumqi 830052, China; pcl109969@163.com; 2Intelligent Equipment Research Center, Beijing Academy of Agriculture and Forestry Sciences, Beijing 100097, China; gzwneau@163.com (Z.G.); wulikh@shu.edu.cn (Y.M.); liyj@nercita.org.cn (Y.L.); vishkcc@163.com (Y.Z.); glzzz2024@163.com (L.G.); 3Beijing Key Laboratory of Intelligent Equipment Technology for Agriculture, Beijing 100097, China

**Keywords:** tomato, harvesting robot, robotic arm, structural parameter optimization, collision-free

## Abstract

The current harvesting arms used in harvesting robots are developed based on standard products. Due to design constraints, they are unable to effectively avoid obstacles while harvesting tomatoes in tight spaces. To enhance the robot’s capability in obstacle-avoidance picking of tomato bunches with various postures, this study proposes a geometric parameter optimization method for a 7 degree of freedom (DOF) robotic arm. This method ensures that the robot can reach a predetermined workspace with a more compact arm configuration. The optimal picking posture for the end-effector is determined by analyzing the spatial distribution of tomato bunches, the main stem position, and peduncle posture, enabling a quantitative description of the obstacle-avoidance workspace. The denavit–hartenberg (D-H) model of the harvesting arm and the expected collision-free workspace are set as constraints. The compactness of the arm and the accessibility of the harvesting space serve as the optimization objectives. The Non-dominated Sorting Genetic Algorithm II (NSGA-II) multi-objective genetic algorithm is employed to optimize the arm length, and the results were validated through a virtual experiment using workspace traversal. The results indicate that the optimized structure of the tomato harvesting arm is compact, with a reachability of 92.88% in the workspace, based on the collision-free harvesting criteria. This study offers a reference for structural parameter optimization of robotic arms specialized in fruit and vegetable harvesting.

## 1. Introduction

As reported by Tomato News [1], global tomato production in 2021 reached around 189.1 million tons, positioning it as one of the crops with the largest planted area and output globally [2]. Since tomatoes need to be harvested in stages at different ripening periods, a large amount of labor is required for selective harvesting [3]. As a result, high labor needs in tomato production highlight the need for harvesting robots. Harvesting robots allow 24 h continuous operation, improving efficiency and reducing reliance on seasonal labor [4,5]. Modern harvesting robots use advanced sensors and vision systems to accurately detect and harvest ripe tomatoes, ensuring fruit quality [6]. Although the initial investment is high, harvesting robots will significantly reduce costs and improve economic efficiency in the long term [7].

Numerous institutions and researchers have conducted extensive studies on tomato harvesting robots, yielding notable results. The company RooT AI developed a small greenhouse-tomato-harvesting robot called Virgo [8]. The robot employs computer vision and motion sensors for navigation, enabling it to avoid vines and unripe fruits. It utilizes cameras and infrared lasers for 3D color scanning to assess ripeness, ensuring that picking occurs without damaging the fruit. According to the current development of the harvesting robot, articulated robotic arms are widely recognized as suitable for tomato harvesting, with a harvesting efficiency of 8–10 s per cluster and a success rate of 75–85% [9,10]. Despite significant advances in harvesting efficiency and accuracy, insufficient obstacle avoidance capability remains a major constraint on operational performance [11,12,13]. Consequently, obstacle avoidance technology has become a critical research focus for harvesting robots. To address this issue, researchers have refined algorithms, integrated advanced sensors, and improved path planning to guide robotic arms in effectively avoiding obstacles during harvesting. Achieving multi-posture obstacle avoidance notably requires robotic arms with high degrees of freedom [14,15]. In practice, high degree of freedom (DOF) robotic arms provide greater flexibility in navigating complex agricultural environments, ensuring efficient and precise harvesting [16]. The robotic arms referenced in the above article are mostly industrial ones. Due to design constraints, they face challenges such as low reachability in harvesting areas, bulky structures, and limited arm span, which hinder efficient operation in tight spaces [17].

The harvesting arm is an automated device developed specifically for the agriculture industry, designed for efficient and damage-free harvesting of fruits and vegetables [18]. As a crucial component of fruit and vegetable harvesting robots, it ensures collision-free harvesting through visual sensing, motion control, and gripper design. Collision-free harvesting refers to the end of the harvesting arm performing the harvesting task without making contact with the crop body, branches, leaves, or other parts of the crops. In this study, collision-free harvesting means the arm can pick tomato bunches without striking the bunches, main stems, or peduncles [19,20,21]. Therefore, optimizing configuration parameters and designing specialized harvesting arms are crucial to improving the performance of current harvesting robots. To address the aforementioned issues, researchers have optimized the geometric parameters of the robotic arm by analyzing the growth patterns and distribution of fruits and vegetables. The optimization focused on minimizing ineffective workspace, designing a more compact robotic arm structure, and implementing multi-arm cooperative harvesting strategies [22,23,24,25]. As a result, the harvesting range increased to over 90% of the target operational area, while reducing the robotic arm’s structural length and improving end-effector movement speed and precision.

The above studies primarily focused on spatial accessibility and structural compactness in manipulator configuration optimization, paying less attention to obstacle-avoidance requirements. Traditional harvesting methods assume that fruit and vegetable peduncles orient vertically downward, requiring only a horizontal posture for the end-effector during harvesting; this increases the likelihood of collisions. The high cellulose and lignin content in tomato peduncles imparts rigidity and tensile strength, resulting in varied peduncle postures. In this study, the end-effector harvesting posture of the robotic arm was determined based on the growth orientation of the main tomato stem and inflorescence peduncle, thereby achieving collision-free harvesting of tomato clusters. This approach prevents collisions with the tomato bunches, main stem, or peduncle, achieving obstacle avoidance during harvesting. Consequently, this increases the demands on the robotic arm.

To improve the robot’s obstacle-avoidance capability when picking tomato clusters with various orientations, this study adopts collision-free harvesting as the premise, focusing on the 7-DOF harvesting robotic arm as the research subject, and proposes a geometric parameter optimization method specifically designed for this type. The goal is to ensure the robot reaches the intended obstacle-avoidance harvesting area while maintaining a compact harvesting arm structure. The main contents of this study are as follows:(1)Based on the industrial greenhouse tomato cultivation model, the spatial distribution characteristics of tomato bunches–peduncles–main stems in random postures are analyzed. A method to describe the obstacle-avoidance picking posture of tomato clusters is proposed, and the target space for collision-free picking is defined.(2)Based on the definition of collision-free harvesting, aiming to maximize the compactness of the harvesting arm structure and the reachable rate of the target picking space, and using the denavit–hartenberg (D-H) kinematic model and target picking space as constraints, the NSGA-II multi-objective genetic algorithm is employed to optimize the harvesting arm length parameters.(3)The potential workspace was traversed through simulation experiments, and the collision-free harvesting reachability of the robotic arm configuration was determined in the simulated environment, confirming the feasibility and effectiveness of the proposed optimization method.

The structure of this paper is as follows: Section 2 proposes a method to describe the obstacle-avoidance picking posture, based on the growth morphology of tomato bunches-peduncles-main stems, and determines the target harvesting workspace. Next, it introduces the robot configuration and establishes the kinematic model of the harvesting arm. Finally, the NSGA-II algorithm is applied to optimize the four arm length parameters of the harvesting arm. Section 3 validates the optimization method through a virtual experiment using workspace traversal, confirming its feasibility and effectiveness. In Section 4, a summary of the paper is provided.

## 2. Materials and Methods 

### 2.1. The Tomato Harvesting Environment and Operation Principle

#### 2.1.1. Industrial Greenhouse Tomato Environment

Industrial greenhouse cultivation is a key method in modern tomato production, with precise control of environmental factors to ensure optimal tomato growth [26]. This method not only improves tomato yield and quality but also provides the foundation for automated harvesting, advancing the modernization and automation of agricultural production. As shown in Figure 1a, the main stem of greenhouse-grown tomato plants grows between 800 mm and 2500 mm above the ground, covering a total height of 1700 mm. Dynamic vine lowering positions ripe tomatoes at a height of 1150 to 1630 mm above the ground, covering a range of 480 mm, to make harvesting easier. The tomato fruits have a thickness of 200 mm, with a center-to-center distance of 1200 mm between ripe tomatoes on either side. The main stem is angled between 30° and 60° relative to the ground.

#### 2.1.2. Collision-Free Harvesting Operation Requirements

The tomato is a herbaceous plant; tomato bunches are connected to the main stem through the peduncle, as shown in Figure 1b. As a living organism, the growth posture of tomatoes exhibits a random distribution. In this study, tomato growth postures were categorized into four orientations—front, back, left, and right—based on the peduncle’s orientation, as shown in Figure 2a–d. Additionally, an analysis was conducted on the distribution characteristics of 378 tomato bunches from multiple rows in the greenhouse.

The results, shown in Table 1, indicate that the slanted growth of the main stem and the weight of the fruit caused tomatoes to be predominantly distributed on the right side, followed by the front and back, with the fewest tomatoes on the left. Based on the sample data, a distribution range of peduncle orientations was plotted in Figure 3a. The transparent gray, pink, blue, and red areas indicate four regions corresponding to the peduncle orientations: front (I), back (II), left (III), and right (IV). 

As shown in Figure 2e, during manual tomato picking, to achieve obstacle avoidance for tomato bunches with random postures, the left hand is first used to lift the tomato bunch, exposing the peduncle fully within the field of view, and the right hand then uses scissors to perform the cutting operation. During picking, the distance between the scissors and the intersection of the main stem and peduncle is maintained between 5 mm and 15 mm, with the scissors positioned as perpendicular as possible to the main stem and peduncle, to avoid collisions with the stem and fruit. However, when the robot is used for picking, the inability to perform the lifting action limits the visibility of the peduncle in the field of view, limiting the length of the peduncle that can be cut. Therefore, the picking process requires more precise operations to ensure the fruit remains undamaged.

#### 2.1.3. Collision-Free Harvesting Posture Description for the Peduncle-Cutting End-Effector

During harvesting operations, the robot must determine both the position and orientation of the peduncle. Therefore, introducing the concept of a peduncle direction vector is essential. The picking posture and the position of the picking point are determined by combining the directions of the main stem and peduncle. As shown in Figure 3a, point $A(x_{1},y_{1},z_{1})$ is located at the intersection of the main stem and peduncle. Point $B(x_{2},y_{2},z_{2})$ is positioned 30 mm upwards along the main stem, point $O(x_{0},y_{0},z_{0})$ is placed 10 mm along the peduncle, and point $C(x_{3},y_{3},z_{3})$ is 20 mm further along the peduncle. Then, vector ${\vec{l}}_{AB}$ is established from point *A* to point *B*, and vector ${\vec{l}}_{OC}$ from point *O* to point *C*. Finally, point *O* is set as the picking point, with ${\vec{l}}_{AB}$ as the main stem vector and ${\vec{l}}_{OC}$ as the peduncle vector, used to describe the end-effector’s pose, leading to the desired end-effector homogeneous pose matrix for the robotic arm.

As illustrated in Figure 4, the peduncle cutting end-effector is a single-drive dual-action mechanism consisting mainly of a fixed base, cylinder, sliding block, fixing block, shears and clamp. The fixing block is mounted on the fixed base, and the cylinder shaft is firmly attached to the sliding block. The sliding block is connected to the shears and clamp via side pins. During operation, the cylinder drives the sliding block forward, causing the shears and clamp to rotate around the central pin, completing the cutting and clamping tasks.

To improve the picking reachability rate and prevent collisions between the gripper’s end and the main stem or fruit, the ideal picking posture for the end-effector is established based on the manual picking process described in Section 2.1.2. As illustrated in Figure 3b, the ideal collision-free picking posture is achieved when plane U, formed by the *Z*-axis of the end-effector coordinate system and vectors ${\vec{l}}_{AB}$ and ${\vec{l}}_{OC}$, is vertical, and the end-effector’s *X*-axis coincides with ${\vec{l}}_{OC}$. 

Given that tomato bunches and peduncles tend to grow in the direction of gravity, and based on practical experience from manual picking, when the angle between the *Z*-axis of the end-effector and the peduncle vector is within 60° to 120°, the likelihood of interference and collisions is reduced. Therefore, this study sets the angle between the end-effector’s *Z*-axis and the peduncle vector ${\vec{l}}_{OC}$ to be within the 60–120° range, deeming this posture a prerequisite for safe picking; in other words, this is what collision-free harvesting means.

The end-effector attitude matrix is used to describe the posture of the robot’s peduncle cutting end-effector. To define the desired posture of the manipulator’s end-effector, the three axial unit vectors of the end-effector’s coordinate system {E} (*X*-axis, *Y*-axis, and *Z*-axis) relative to the manipulator’s base coordinate system {B} must be derived [27]. Using this approach, the spatial orientation and position of the peduncle cutting end-effector can be clearly defined. The detailed steps are as follows:

Let the unit vector of the *X*-axis of the end-effector coordinate system be denoted as ${\vec{X}}_{e}(n_{x},n_{y},n_{z})$, the unit vector of the *Y*-axis as ${\vec{Y}}_{e}(o_{x},o_{y},o_{z})$, and the unit vector of the *Z*-axis as ${\vec{Z}}_{e}(a_{x},a_{y},a_{z})$. Based on the previously defined ideal picking posture, the following can be established:$${\vec{Z}}_{e}=\frac{{\vec{l}}_{AB}\times {\vec{l}}_{OC}}{\left\|{\vec{l}}_{AB}\times {\vec{l}}_{OC}\right\|},\;{\vec{X}}_{e}=\frac{{\vec{l}}_{OC}}{\left\|{\vec{l}}_{OC}\right\|}$$

Next, based on the definition of the end-effector attitude matrix (where the *X*-axis, *Y*-axis, and *Z*-axis are unit vectors and mutually perpendicular), the unit vector ${\vec{Y}}_{e}={\vec{X}}_{e}\times {\vec{Z}}_{e}$ for the *Y*-axis of the end-effector is calculated [28]. The ideal picking posture matrix *R_s_* is created using the ${\vec{X}}_{e}$, ${\vec{Y}}_{e}$, and ${\vec{Z}}_{e}$ coordinates, as shown in Equation (1), and set as the initial end-effector pose matrix.
(1)$$R_{s}=\left[\begin{array}{ccc} n_{x} & o_{x} & a_{x}\\ n_{y} & o_{y} & a_{y}\\ n_{z} & o_{z} & a_{z} \end{array}\right]$$

After acquiring the initial end-effector attitude matrix for the manipulator, by rotating around the end-effector coordinate system *Y*-axis, the corresponding attitude matrix in the range of 60–120° is generated [29]. The specific method is to multiply the initial attitude matrix *R_s_* with the rotation matrix Rot(y, *θ*) rotating about the Ye-axis. The formula is as follows:(2)$$\begin{array}{cc} R_{P} & =R_{s}Rot(y,\varepsilon )\\ & \begin{array}{c} =\left[\begin{array}{ccc} n_{x} & o_{x} & a_{x}\\ n_{y} & o_{y} & a_{y}\\ n_{z} & o_{z} & a_{z} \end{array}\right] \end{array}\left[\begin{array}{ccc} \cos\varepsilon & 0 & \sin\varepsilon \\ 0 & 1 & 0\\ -\sin\varepsilon & 0 & \cos\varepsilon \end{array}\right] \end{array}$$
where $\varepsilon =\frac{\pi }{2}-\alpha$.

Finally, using point $O(x_{0},y_{0},z_{0})$ as the picking point, the acceptable posture is taken as the final picking posture. The end-effector homogeneous pose matrix for the collision-free picking posture, as shown in Equation (3), is constructed.
(3)$$T_{p}=\left[\begin{array}{cccc} n_{x}\cos\varepsilon -a_{x}\sin\varepsilon & o_{x} & n_{x}\sin\varepsilon +a_{x}\cos\varepsilon & x_{0}\\ n_{y}\cos\varepsilon -a_{y}\sin\varepsilon & o_{y} & n_{y}\sin\varepsilon +a_{y}\cos\varepsilon & y_{0}\\ n_{z}\cos\varepsilon -a_{z}\sin\varepsilon & o_{z} & n_{z}\sin\varepsilon +a_{z}\cos\varepsilon & z_{0}\\ 0 & 0 & 0 & 1 \end{array}\right]$$
where
$$\begin{array}{l} o_{x}=(((\frac{z_{2}-z_{1}}{\left\|{\vec{l}}_{AB}\right\|}\cdot \frac{x_{3}-x_{0}}{\left\|{\vec{l}}_{OC}\right\|})-(\frac{x_{2}-x_{1}}{\left\|{\vec{l}}_{AB}\right\|}\cdot \frac{z_{3}-z_{0}}{\left\|{\vec{l}}_{OC}\right\|}))\cdot \frac{z_{3}-z_{0}}{\left\|{\vec{l}}_{OC}\right\|})-(((\frac{y_{2}-y_{1}}{\left\|{\vec{l}}_{AB}\right\|}\cdot \frac{z_{3}-z_{0}}{\left\|{\vec{l}}_{OC}\right\|})-(\frac{z_{2}-z_{1}}{\left\|{\vec{l}}_{AB}\right\|}\cdot \frac{y_{3}-y_{0}}{\left\|{\vec{l}}_{OC}\right\|}))\cdot \frac{y_{3}-y_{0}}{\left\|{\vec{l}}_{OC}\right\|});\\ o_{y}=(((\frac{z_{2}-z_{1}}{\left\|{\vec{l}}_{AB}\right\|}\cdot \frac{x_{3}-x_{0}}{\left\|{\vec{l}}_{OC}\right\|})-(\frac{x_{2}-x_{1}}{\left\|{\vec{l}}_{AB}\right\|}\cdot \frac{z_{3}-z_{0}}{\left\|{\vec{l}}_{OC}\right\|}))\cdot \frac{x_{3}-x_{0}}{\left\|{\vec{l}}_{OC}\right\|})-(((\frac{y_{2}-y_{1}}{\left\|{\vec{l}}_{AB}\right\|}\cdot \frac{z_{3}-z_{0}}{\left\|{\vec{l}}_{OC}\right\|})-(\frac{z_{2}-z_{1}}{\left\|{\vec{l}}_{AB}\right\|}\cdot \frac{y_{3}-y_{0}}{\left\|{\vec{l}}_{OC}\right\|}))\cdot \frac{z_{3}-z_{0}}{\left\|{\vec{l}}_{OC}\right\|});\\ o_{z}=(((\frac{x_{2}-x_{1}}{\left\|{\vec{l}}_{AB}\right\|}\cdot \frac{y_{3}-y_{0}}{\left\|{\vec{l}}_{OC}\right\|})-(\frac{y_{2}-y_{1}}{\left\|{\vec{l}}_{AB}\right\|}\cdot \frac{x_{3}-x_{0}}{\left\|{\vec{l}}_{OC}\right\|}))\cdot \frac{y_{3}-y_{0}}{\left\|{\vec{l}}_{OC}\right\|})-(((\frac{y_{2}-y_{1}}{\left\|{\vec{l}}_{AB}\right\|}\cdot \frac{z_{3}-z_{0}}{\left\|{\vec{l}}_{OC}\right\|})-(\frac{z_{2}-z_{1}}{\left\|{\vec{l}}_{AB}\right\|}\cdot \frac{y_{3}-y_{0}}{\left\|{\vec{l}}_{OC}\right\|}))\cdot \frac{x_{3}-x_{0}}{\left\|{\vec{l}}_{OC}\right\|});\\ a_{x}=(\frac{y_{2}-y_{1}}{\left\|{\vec{l}}_{AB}\right\|}\cdot \frac{z_{3}-z_{0}}{\left\|{\vec{l}}_{OC}\right\|})-(\frac{z_{2}-z_{1}}{\left\|{\vec{l}}_{AB}\right\|}\cdot \frac{y_{3}-y_{0}}{\left\|{\vec{l}}_{OC}\right\|});\;a_{y}=(\frac{z_{2}-z_{1}}{\left\|{\vec{l}}_{AB}\right\|}\cdot \frac{x_{3}-x_{0}}{\left\|{\vec{l}}_{OC}\right\|})-(\frac{x_{2}-x_{1}}{\left\|{\vec{l}}_{AB}\right\|}\cdot \frac{z_{3}-z_{0}}{\left\|{\vec{l}}_{OC}\right\|});\\ a_{z}=(\frac{x_{2}-x_{1}}{\left\|{\vec{l}}_{AB}\right\|}\cdot \frac{y_{3}-y_{0}}{\left\|{\vec{l}}_{OC}\right\|})-(\frac{y_{2}-y_{1}}{\left\|{\vec{l}}_{AB}\right\|}\cdot \frac{x_{3}-x_{0}}{\left\|{\vec{l}}_{OC}\right\|});\;n_{x}=\frac{x_{3}-x_{0}}{\left\|{\vec{l}}_{OC}\right\|};\;n_{y}=\frac{y_{3}-y_{0}}{\left\|{\vec{l}}_{OC}\right\|};\;n_{z}=\frac{z_{3}-z_{0}}{\left\|{\vec{l}}_{OC}\right\|};\\ \left\|{\vec{l}}_{AB}\right\|=\sqrt{{\left(x_{2}-x_{1}\right)}^{2}+{\left(y_{2}-y_{1}\right)}^{2}+{\left(z_{2}-z_{1}\right)}^{2}};\;\left\|{\vec{l}}_{OC}\right\|=\sqrt{{\left(x_{3}-x_{0}\right)}^{2}+{\left(y_{3}-y_{0}\right)}^{2}+{\left(z_{3}-z_{0}\right)}^{2}}. \end{array}$$

### 2.2. Kinematic Model of the Harvesting Arm

#### 2.2.1. The 7-DOF Harvesting Arm

This robot was designed for harvesting tomato bunches, aiming to complete the task efficiently and without damage. As shown in Figure 5a, the robot primarily consists of a rail-guided mobile platform, a fruit box palletizing system, a control system, a lateral slide table, a peduncle cutting end-effector, an Intel RealSense imaging device, and a 7-DOF robotic arm. It is an automated tomato harvesting robot capable of autonomously picking, collecting, and stacking tomato bunches.

Considering the need for obstacle-avoidance picking, a 7-DOF serial robotic arm, with the same degrees of freedom as a human arm, was selected as the harvesting arm, as shown in Figure 5b. The robotic arm features an anthropomorphic design, which gives it higher flexibility, allowing it to rotate at any angle even in confined spaces, thereby significantly improving its operational capabilities [30]. Additionally, this design provides a larger range of motion, enabling the robotic arm to handle more complex tasks and environments. When the robot is located in the greenhouse harvesting environment, the lateral slide table moves to the robot’s central axis, with the robotic arm base mounted on it. The vertical distance from the base’s bottom to the ground is 653 mm, and the horizontal distance between the base’s central axis and the central axes of the mature tomato zones on both sides is 600 mm. The base coordinate system of the robotic arm *O* (*x*_0_, *y*_0_, *z*_0_) is set up on this basis.

The modified D-H method is employed to establish the joint coordinate systems of the robotic arm, as shown in Figure 5b. After establishing the D-H coordinate systems, the link length *a*_*i*_, link offset *d*_*i*_, link twist angle α_*i*_, and joint angle *θ*_*i*_ are determined based on the transformation relationships between adjacent joint coordinate systems [31]. The D-H parameters are shown in Table 2.

The joint rotation angles are as follows:(4)$$\left\{\begin{array}{l} -170{}^{\circ}\leq \theta_{1}\leq 170{}^{\circ}\\ -120{}^{\circ}\leq \theta_{2}\leq 120{}^{\circ}\\ -170{}^{\circ}\leq \theta_{3}\leq 170{}^{\circ}\\ -120{}^{\circ}\leq \theta_{4}\leq 120{}^{\circ}\\ \begin{array}{l} -170{}^{\circ}\leq \theta_{5}\leq 170{}^{\circ}\\ -120{}^{\circ}\leq \theta_{6}\leq 120{}^{\circ}\\ -180{}^{\circ}\leq \theta_{7}\leq 180{}^{\circ} \end{array} \end{array}\right.$$

Subsequently, the modified D-H method is used to establish the forward kinematic model of the robotic arm, with the transformation matrix between adjacent coordinate systems given by the following equation:Tii−1=Rot(x,αi−1)Trans(x,αi−1)Rot(z,θi)Trans(z,di)
(5)Tii−1=cosθi−sinθi0ai−1sinθicosαi−1cosθicosαi−1−sinαi−1−sinαi−1disinθisinαi−1cosθisinαi−1cosαi−1cosαi−1di0001

By substituting the D-H parameters from Table 2 into Equation (5), the transformation matrices for each link of the robotic arm are calculated. Subsequently, by sequentially multiplying these matrices using the right multiplication rule, the transformation matrix of the end-effector coordinate system {E} relative to the base coordinate system {B} is obtained, as shown in Equation (6).
(6)T70=10T21T32T43T54T65T76T=nxoxaxpxnyoyaypynzozazpz0001

#### 2.2.2. Inverse Kinematics Solution of the Harvesting Arm

The inverse kinematics problem is a nonlinear issue that involves redundancy, multiple variables, and strong coupling. Common approaches to solving this include graphical, analytical, and numerical methods [32]. Graphical methods are useful for intuitive problems in simple robotic arms, while analytical methods are suited for arms with explicit mathematical models and simple constraints. Numerical methods are more appropriate for complex mechanical structures with kinematic constraints, especially when high-precision solutions or real-time computations are required. Since the inverse kinematics of a 7-DOF robotic arm typically exhibits nonlinear characteristics and multiple solutions, this study employs the particle swarm optimization (PSO) algorithm to solve the inverse kinematics problem.

The PSO algorithm simulates the movement and evolution of particles in the solution space, gradually converging to the optimal solution. Each particle represents a candidate solution and has two key properties: position and velocity. In PSO, each particle’s velocity is updated based on both its historically best position and the global best position of the swarm [33]. The velocity and position update equations for the inverse kinematics of the robotic arm are as follows:(7)$$u_{i,j}(t+1)=\lambda \cdot u_{i}(t)+c_{1}\cdot r_{1}\cdot (p_{i,j}-x_{i,j}(t))+c_{2}\cdot r_{2}\cdot (g_{j}-x_{i,j}(t))$$
(8)$$x_{i,j}(t+1)=x_{i,j}(t)+u_{i,j}(t+1)$$
where $u_{i,j}(t)$ denotes the velocity of particle *i* in the *j*-th dimension (joint angle) during the *t*-th iteration, i.e., the rate of change in the joint angle. $x_{i,j}(t)$ is the position of particle *i* in the *j*-th dimension (joint angle) during the *t*-th iteration, representing the current joint angle combination. $p_{i,j}$ refers to the historical best position of particle *i* in the *j*-th dimension (joint angle) up to the current iteration, which is the best combination of joint angles found by particle *i* so far. $g_{j}$ is the global best position of the population in the *j*-th dimension (joint angle) up to the current iteration, i.e., the solution closest to the target end-effector position found by the group so far. λ is the inertia weight, controlling how much of the particle’s previous velocity is retained; in this study, λ = 0.5. *c*_1_ and *c*_2_ are acceleration coefficients, representing the weights of personal and social experiences, respectively; in this study, *c*_1_ = 2 and *c*_2_ = 2. *r*_1_ and *r*_2_ are random numbers uniformly distributed in the range [0,1].

To assess the quality of the particle’s current position, a fitness function must be defined. To better adapt to the search mechanism of the algorithm and improve convergence speed, the fitness function in this study utilizes RPY (Roll, Pitch, and Yaw) [28] angles to convert the 3D matrix into a 3D vector. Using Equation (9), the rotation angles around the *Z*-axis, *Y*-axis, and *X*-axis of the base coordinate system {B}—α(roll), β(pitch), and γ(yaw)—are computed. Then, the target posture vector ${\vec{a}}_{1}(\alpha_{1},\beta_{1},\gamma_{1})$ and the desired posture vector ${\vec{a}}_{0}(\alpha_{0},\beta_{0},\gamma_{0})$ are constructed using these angles. The target posture vector represents the specific position and orientation of the robotic arm’s end-effector as required by the task, which is the final posture the arm must achieve. The desired posture vector refers to the temporary posture expected by the arm, based on its current state and motion planning; it dynamically and incrementally approaches the target posture. Similarly, the target and desired position vectors discussed below follow the same concept.
(9)$$\left\{\begin{array}{l} \beta =A\tan2(-r_{31},\sqrt{r_{11}^{2}+r_{21}^{2}})\\ \alpha =A\tan2(r_{21},r_{11})\\ \gamma =A\tan2(r_{32},r_{33}) \end{array}\right.$$

In this study, a fitness function is designed to evaluate the quality of the particle’s current solution, considering both position error and posture error of the end-effector. The fitness function is defined using the 2-norm, with the threshold set at 10^−5^. The detailed definition of the function is given in Equation (10).
(10)$$\delta ={\left\|{\vec{b}}_{1}-{\vec{b}}_{0}\right\|}_{2}+{\left\|{\vec{a}}_{1}-{\vec{a}}_{0}\right\|}_{2}=\sqrt{\sum_{i=1}^{3}{\left|{\vec{b}}_{1}(i)-{\vec{b}}_{0}(i)\right|}^{2}}+\sqrt{\sum_{j=1}^{3}{\left|{\vec{a}}_{1}(j)-{\vec{a}}_{0}(j)\right|}^{2}}$$
where ${\vec{b}}_{1}$ is the target position vector; ${\vec{b}}_{0}$ is the desired position vector; and (*i*) and (*j*) represent individual elements of the current vector.

To ensure the validity of the inverse solution, joint angle limits must be imposed, with the specific angle limits provided in Equation (4). Additionally, the desired pose matrix of the robotic arm’s end-effector is calculated, based on a given set of joint angles, using the forward kinematics Equation (6). In this study, the population size is set to 60 particles, and the dimensionality is set to 7, corresponding to the 7 degrees of freedom of the robotic arm. To reasonably determine the maximum number of iterations, the particle swarm optimization algorithm is employed to find inverse solutions for five end-effector poses. The maximum number of iterations is set to 50, 100, 200, 500, 1000, and 5000, respectively. The time consumed and the average fitness values for different end-effector positions are shown in Table 3, all under the same maximum number of iterations.

According to Table 3, the function essentially converges after 200 iterations, with the fitness difference between 200 iterations and those of 500, 1000, and 5000 iterations being less than 10^−9^. However, the time consumption for 500, 1000, and 5000 iterations is 2.5, 5, and 25 times that of 200 iterations, respectively. Therefore, the maximum number of iterations in this study is set to 200.

### 2.3. Structural Parameter Optimization of the Harvesting Arm

#### 2.3.1. Simplified Working Space for Tomato Harvesting

This study aims to optimize the geometric parameters of the robotic arm to enhance picking reachability while ensuring a more compact mechanical structure, providing an efficient, stable, and lightweight specialized arm for tomato harvesting robots. Before establishing the objective function for geometric parameter optimization and determining the optimization conditions, it is first necessary to establish a simplified workspace for the tomato picking task. Based on the analysis of the industrial greenhouse environment and the growth characteristics of tomato plants, the tomato picking task simplified in the workspace is constructed to closely simulate the actual picking process.

This study selects the tomato space on the robot’s left side (along the +Y axis of the robotic arm) as the optimized workspace. The workflow for building the tomato picking model is as follows: (1)Predefined workspace configuration: based on the camera’s field of view, the *X*-axis is set from −200 mm to 200 mm; the *Y*-axis is set from 500 mm to 700 mm according to the mature tomato thickness of 200 mm; and the *Z*-axis is set from 497 mm to 977 mm based on the robot arm’s height of 653 mm above the ground.(2)Main stem vector generation: a random crossing point of the tomato plant is generated within the predefined workspace and set as the origin. Random vectors forming angles of 30° to 60° with the +X axis of the robotic arm are established as the main stem vectors.(3)Peduncle vector generation: starting from the crossing point, 1000 uniformly distributed endpoints of peduncles are established, and each peduncle vector is generated accordingly.(4)Peduncle vector selection: based on the predefined workspace and classification, the 1000 end-points are filtered and categorized. According to the actual growth posture distribution, 16 vectors are selected as the final peduncle vectors: 2 from region I, 2 from region II, 1 from region III, and 11 from region IV (these vectors are randomly selected within their region).(5)Picking point generation: sixteen picking points were established on the peduncle vectors, positioned 10 mm from the origin.

The potential picking tasks for one tomato bunch are shown in Figure 6. Finally, 100 randomized tomato picking models are created in the predefined workspace, with at least 1600 end-effector poses generated for each picking operation, ensuring the robustness of the optimization results.

#### 2.3.2. Arm Length Parameter Optimization

This study evaluates the performance of the robotic arm using two key indicators: picking reachability and the structural parameters of the robotic arm. Using these indicators, objective function 1 and objective function 2 are formulated.

Optimizing arm length can directly and effectively influence the robotic arm’s working range, while maintaining simplicity and stability in its structure. In the D-H parameters of the anthropomorphic robotic arm, the values of *a*_2_, *a*_4_, and *a*_6_ are all zero. Given this configuration’s excellent operational capabilities in narrow spaces, this study does not modify the configuration but focuses on optimizing the four arm length parameters *d*_1_, *d*_3_, *d*_5_, and *d*_7_.

As concluded earlier from manual picking experience, it was observed that when the peduncle cutting end-effector is in the ideal picking posture, the scissors almost never interfere with the main stem. Similarly, in an acceptable picking posture, the probability of collision is also low. In other words, the closer the angle between the *Z*-axis of the end-effector’s coordinate system and the peduncle vector is to 90°, the lower the chance of collision interference and the higher the collision-free picking reachability. Therefore, the angle between the *Z*-axis of the end-effector and the peduncle vector is used to establish the objective function. The specific steps are as follows: 

(1) Obtain the end-effector pose matrix for the picking point using the inverse kinematics solution;

(2) Calculate the angle between the end-effector pose matrix and the peduncle vector; 

(3) Use the formula 90° − α, and take the absolute value to meet the minimization requirement of the algorithm;

(4) Average all the angles to establish objective function 1, as shown in Equation (11).
(11)$$\sigma =\frac{\sum_{n=1}^{J}\left(\frac{\pi }{2}-\left|\alpha_{n}\right|\right)}{J}$$
where *J* represents the set of generated tomato peduncle posture in the simplified picking workspace. In this paper, *J* = 1600; *n* represents the current posture of the tomato peduncle; $\alpha_{n}$ represents the picking angle of the end-effector posture corresponding to each tomato peduncle posture, $\alpha_{n}\in [\pm 90{}^{\circ},0{}^{\circ})$.

The structural parameters of the robotic arm are another key indicator for assessing its performance; objective function 2, used to assess the robotic arm’s structure, is defined as shown in Equation (12).
(12)$$L=d_{1}+d_{3}+d_{5}+d_{7}$$

As can be inferred from the previous section, the angle α depends on the posture of the tomato peduncle and the pose of the peduncle cutting end-effector. In the previous section, the end-effector pose matrix was derived using the robotic arm’s inverse kinematics. It becomes clear that the angle is associated with the variables *d*_1_, *d*_3_, *d*_5_, and *d*_7_ as well as the joint angles. Therefore, Equation (13) is used to define the mathematical relationships among these variables. The range of joint angle *θ*_*i*_ is given in Equation (4).
(13)$$f=\alpha (d_{1},d_{3},d_{5},d_{7},\theta_{i})$$
where *i* = 1–7.

The optimization goal of this study is to maximize picking reachability and achieve the most compact mechanical structure. As objective function 1 approaches 0, α moves closer to 90°, reducing the likelihood of collision interference and improving picking reachability. When *L* decreases, the robotic arm structure becomes more compact. Hence, during optimization, both σ and *L* should be minimized as much as possible.

To ensure that the solution to the objective function remains within a reasonable range, constraints on the parameters are necessary. By imposing constraints on the four arm length parameters, invalid optimization results can be avoided, ensuring a more rigorous optimization process and leading to the optimal solution for the objective function. The main constraints are as follows: 

(1) Joint structure constraint: considering the minimum length of the joint structure, the arm length parameters must be greater than 100 mm. Since the end-effector is 175 mm, *d*_7_ must exceed 275 mm. 

(2) Spatial constraint: to prevent the robotic arm from colliding with nearby tomato vines or the robot during picking, *d*_1_ and *d*_5_ should be less than 600 mm, and *d*_3_ should be less than 500 mm. 

(3) Clearance constraint: the minimum distance between tomato main stems is 300 mm; to avoid interference when picking tomatoes behind the main stem, *d*_7_ must be less than the distance between main stems.

In summary, the geometric parameter optimization function model for the robotic arm is as follows:(14)$$\left\{\begin{array}{l} \min\sigma =\frac{\sum_{n=1}^{J}\left(\frac{\pi }{2}-\left|\alpha_{n}\right|\right)}{J}\\ \min L=d_{1}+d_{3}+d_{5}+d_{7}\\ s.t\\ 100\leq d_{1},d_{5}\leq 600\\ 100\leq d_{3}\leq 500\\ 275\leq d_{7}\leq 300 \end{array}\right.$$

In this study, the multi-objective genetic algorithm NSGA-II (Non-dominated Sorting Genetic Algorithm II) is employed to optimize the analytical expressions of the two objective functions [34]. NSGA-II is a genetic algorithm designed to solve multi-objective optimization problems by using fast non-dominated sorting and an elitist strategy. The steps are as follows:

(1) Generate an initial population consisting of a given number of individuals, and then conduct fast non-dominated sorting on this population.

(2) Perform binary selection to choose individuals from the initial population for crossover and mutation, generating a new population.

(3) Merge the newly generated population with the initial population, perform non-dominated sorting on all individuals, and compute crowding distance within the non-dominated set.

(4) Select the next-generation population, prioritizing individuals with lower non-dominated ranks. For individuals of the same rank, those with higher crowding distance are selected first to ensure diversity.

(5) If the number of generations reaches 10 or the convergence coefficient reaches 10^−6^, the algorithm terminates, and the current population becomes the optimal Pareto set. Otherwise, return to step (2) to continue the iteration.

Following the above steps, the optimal individual coefficient *λ* = 0.3, the population size *N* = 300, the maximum iteration count *G* = 10, the crossover probability *p_c_* = 0.9, the mutation probability *p_m_* = 0.1, and the convergence coefficient *β* = 10^−6^. The initial parameters for the robotic arm are *d*_1_ = 341.5 mm, *d*_3_ = 394 mm, *d*_5_ = 366 mm, and *d*_7_ = 400.3 mm for optimization. The optimization results show that the optimal values are *d*_1_ = 122.3 mm, *d*_3_ = 363.3 mm, *d*_5_ = 600 mm, and *d*_7_ = 287 mm, with the objective function values being σ= 6.275, *L* = 403.3507.

The geometric parameter optimization method for the 7-DOF robotic arm based on NSGA-II is described in Algorithm 1.
**Algorithm 1** The geometric parameter optimization method for the 7-DOF robotic arm based on NSGA-II1. Set algorithm parameters, population size *N* = 300, best individual coefficient *λ* = 0.3, crossover probability *p_c_* = 0.9, mutation probability *p_m_* = 0.1, convergence threshold *β* = 10^−6^, maximum iterations *G* = 10, best *fitness best_fitness* = ∞.2. for iteration count *t* = 1 to *G* do:3.   Select individuals based on fitness, generate new individuals using pc, randomly adjust new individuals using pm, merge parent and offspring populations: *R* ← *P + Q*.4.   Perform non-dominated sorting on population *R* based on multiple objectives.5.   Initialize new population *P = [di]* (a set of candidate solutions, each individual represents a configuration of the robotic arm).6.   while *len(P) + len(F[i]) ≤ N* do:7.       Add individuals from front *F[i]* to population *P*.8.       Increase index: *i = i + 1*.9.    end while.10.   for each individual in *P* do:11.      Build the robotic arm model and tomato model.12.      for peduncle posture *n = 1* to *J* do (*n* represents the current peduncle posture, *J* represents the total number of peduncle postures):13.         while picking angle α ≠ 0° do:14.             Initialize picking angle α = 90° and *k = 1*.15.             Solve inverse kinematics, test the reachability of the robotic arm for the current peduncle posture.16.             if reachable then: record successful angle *α_n_ = α*, and exit loop.17.             else: adjust angle and reverse direction *a = k × (a − 2), k = k × (−1)*.18.             end if.19.         end while.20.      end for.21.      Calculate objective function 1: *σ* (Equation (11)), and objective function 2: *L* (Equation (12))22.      Calculate the fitness of the current individual: *individual.fitness = min(σ, L)*.23.   end for.24.   Calculate the best fitness of the current population: *current_best = calculate_best_fitness(P)*.25.   if *abs(current_best − best_fitness) < β* then: return *P*.26.      Update best fitness: *best_fitness = current_best*.27.   end if.28. end for.29. Return: Final Pareto front solution set *P*.

## 3. Results and Discussion

### 3.1. Analysis of Optimal Solution Validity

Based on the description of collision-free harvesting, this study defines picking reachability as: if the angle φ between the end-effector’s *Z*-axis and the peduncle vector is between 60° and 120°, the robotic arm is considered reachable at that point; refer to Equation (15) for the specific calculation formula. Otherwise, the point is considered inaccessible.
(15)$$\varphi =\cos^{-1}(\frac{{\vec{Z}}_{e}\cdot {\vec{l}}_{oc}}{\left|{\vec{Z}}_{e}\right|\cdot \left|{\vec{l}}_{oc}\right|})$$

To validate the reasonableness and uniqueness of the optimal solution, it is necessary to analyze the effectiveness of the optimal arm length parameters. In this study, the multi-parameter optimization problem is converted into a single-parameter analysis [35,36]. The method is as follows: using picking reachability as the evaluation criterion, one arm length parameter is selected for linear variation, while the other three arm length parameters are constrained to constant values, and the optimal value of the variable parameter is determined. By applying this method to each arm length parameter, the optimal values are obtained and compared with the original optimal solution for validation.

As shown in Figure 7, *d*_1_ and *d*_7_ have minimal impact on picking reachability, while *d*_3_ and *d*_5_ are the primary factors influencing it. In Figure 7a, the range of the picking reachability curve is 5.2%, with minimal variation in the curve. When *d*_1_ = 122.3 mm, the picking reachability reaches its peak at 93.02%, after which the curve levels off. Figure 7b shows that as *d*_3_ increases, picking reachability continuously improves until *d*_3_ = 363.3 mm, where it peaks at 93.22%, after which the curve flattens out. Figure 7c illustrates a pattern similar to *d*_3_ for *d*_5_, with the picking reachability converging to 95.23% when *d*_5_ = 623.2 mm, but since this exceeds the constraint in Equation (14), *d*_5_ is set to 600 mm, resulting in a reachability of 92.88%. Figure 7d shows that the range of the picking reachability curve is 2.1%, with the curve remaining nearly unchanged within the interval. The picking reachability reaches its highest value of 93.29% when *d*_7_ = 287 mm. In conclusion, when *d*_1_ = 122.3 mm, *d*_3_ = 363.3 mm, *d*_5_ = 600 mm, and *d*_7_ = 287 mm, the robotic arm achieves the highest picking reachability in the workspace. This is consistent with the previously optimized parameters, and under these conditions, the picking reachability is at least 92.88%.

### 3.2. Analysis of Obstacle-Avoidance Harvesting Reachability

The peduncle vectors generated by different spatial points vary, resulting in different end-effector poses. To verify whether the end-effector pose matrices obtained through this method can meet the picking requirements and achieve the intended goals, an analysis of the reachability at spatial points is necessary. As shown in Figure 8, 12 specific points in the picking space were selected for picking analysis. At each selected spatial point, 3000 uniformly distributed vectors were generated. After filtering out unreasonable vectors, simulated picking was performed using the acceptable posture of the robotic arm. According to the definitions of accessible and inaccessible picking mentioned earlier, the reachability at spatial points is illustrated in Figure 8.

By applying the regional division method described in Section 2.1.2, as shown in Figure 8, when the peduncle vector points are located at the top of the left region (Region III) and the bottom of the right region (Region IV), the picking reachability decreases significantly. These two areas are challenging for the robotic arm to reach given the current parameters. A comprehensive analysis of the picking performance at different spatial points shows that the orientation of the peduncle vector significantly impacts picking reachability. When the peduncle vector points are located behind the main stem (Region II), picking reachability is the highest; in the front (Region I), it is slightly lower; in the right region (Region IV), it decreases further; and in the left region (Region III), it is the lowest. Further analysis reveals that points A2, A6, and A10 have extremely low picking reachability, as these points are positioned at the rear-top of the workspace. However, points A3, A7, and A11, as well as A1, A5, and A9 located similarly in the rear-top, show higher picking reachability. This indicates significant differences in picking performance depending on the spatial location of the picking point. Therefore, a more detailed analysis of reachability at various locations in the workspace is required.

The reachability among the workspace presents the simulation results for over 12,000 picking points, where color changes represent picking reachability. As shown in Figure 9, most regions exhibit a reachability above 83%, some regions fall between 49% and 83%, and only a few regions have a reachability between 32% and 49%. Moreover, the figure shows that as the X, Y, and Z coordinates increase, the picking reachability gradually declines. This suggests that when picking points are located in the upper-rear area of the workspace, picking difficulty increases, resulting in a sharp reduction in reachability.

In summary, the optimized arm length parameters and the picking reachability of the robotic arm are listed in Table 4. The total length of the parameters is 1372.9 mm, and the picking reachability in the workspace is 92.88%.

## 4. Conclusions

To meet the requirements for collision-free operation of tomato harvesting robots, this study proposes a safe harvesting posture description method for the natural growth morphology of the tomato bunch–peduncle–main stem, drawing on manual operation guidelines. A geometric parameter optimization method for a 7-DOF harvesting arm is developed, focusing on arm compactness and collision-free reachability. The optimal key arm lengths obtained are *d*_1_ = 122.3 mm, *d*_3_ = 363.3 mm, *d*_5_ = 600 mm, and *d*_7_ = 287 mm, achieving a collision-free reachability of 92.88% in the harvesting workspace, which meets the practical requirements of robotic harvesting. This research can serve as a reference for developing specialized robotic arms for fruit and vegetable harvesting.

## Figures and Tables

**Figure 1 plants-13-03211-f001:**
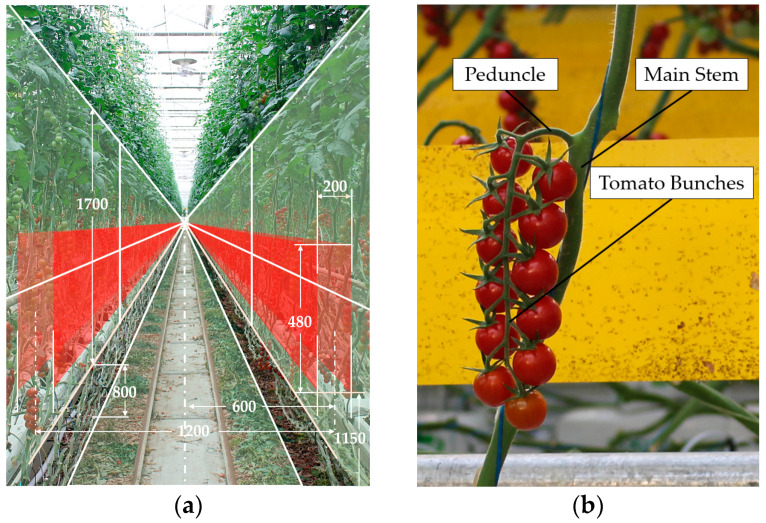
Industrial greenhouse tomato environment. (**a**) Workspace of tomato harvesting; (**b**) tomato bunch on plant.

**Figure 2 plants-13-03211-f002:**
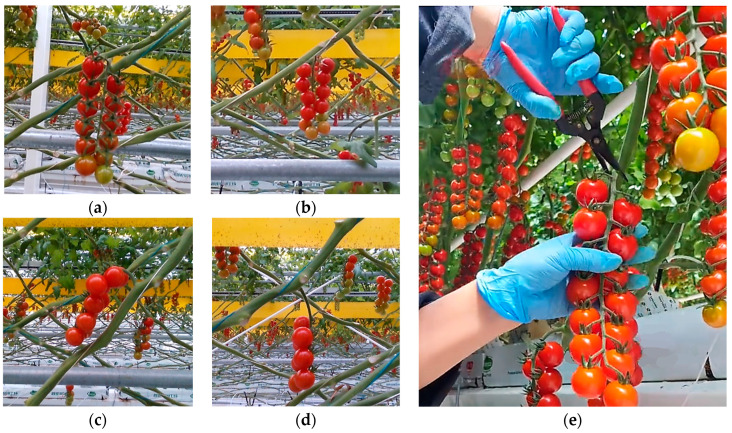
Growth morphology of tomato bunches and requirements for obstacle-avoidance picking operations. (**a**) Tomato bunches with peduncles oriented in front of the main stem; (**b**) tomato bunches with peduncles oriented to the back of the main stem; (**c**) tomato bunches with peduncles oriented to the left of the main stem; (**d**) tomato bunches with peduncles oriented to the right of the main stem; (**e**) manual method for picking tomato bunches.

**Figure 3 plants-13-03211-f003:**
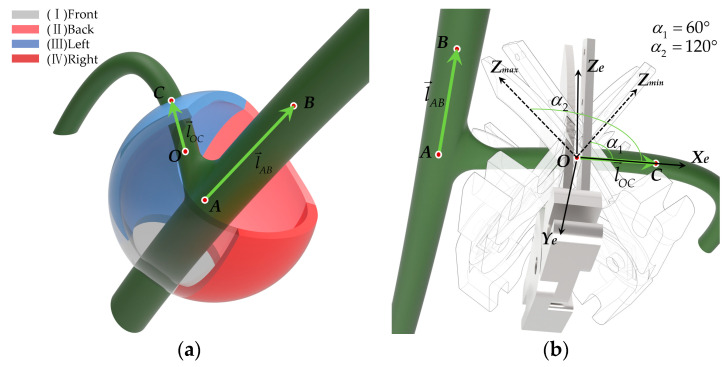
Tomato characteristics and picking posture description. (**a**) Tomato peduncle orientation distribution area and peduncle posture description; (**b**) description of acceptable picking posture.

**Figure 4 plants-13-03211-f004:**
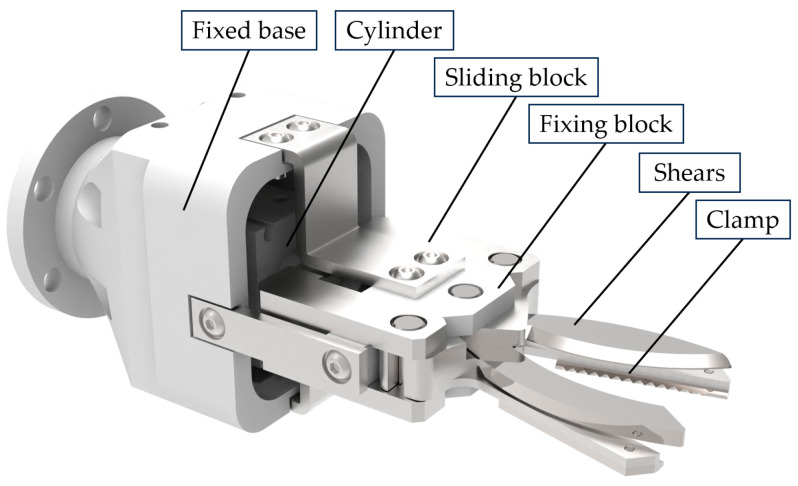
Peduncle cutting end-effector.

**Figure 5 plants-13-03211-f005:**
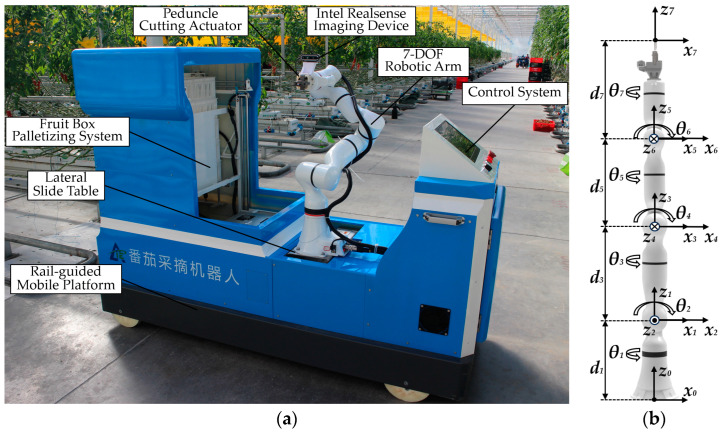
Tomato harvesting robot model. (**a**) Robot model and its components; (**b**) 7-DOF robotic arm model and its coordinate system.

**Figure 6 plants-13-03211-f006:**
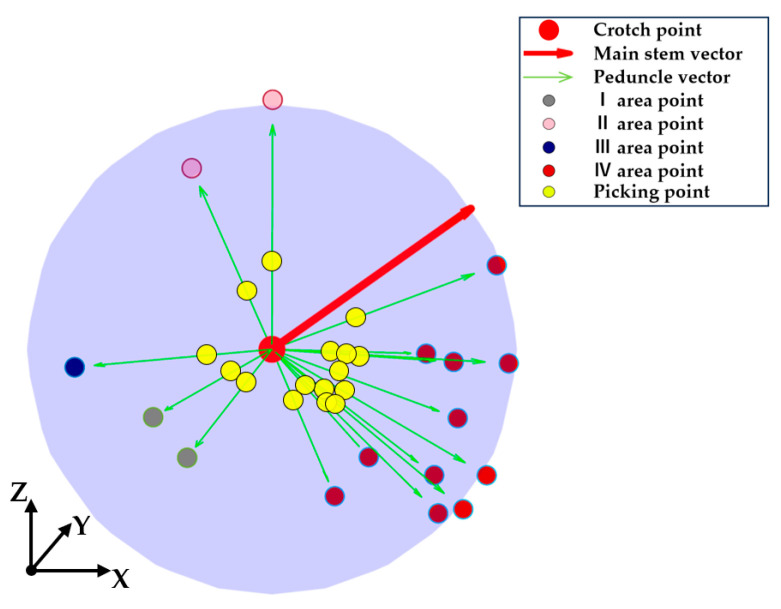
Potential picking tasks for one tomato bunch. The red point indicates the intersection point between the peduncle and the main stem, the red arrow indicates the main stem vector, the green arrow indicates the peduncle vector, the yellow point indicates the picking point, and the colors gray, pink, blue, and light red, respectively, indicate random points within regions I, II, III, and IV.

**Figure 7 plants-13-03211-f007:**
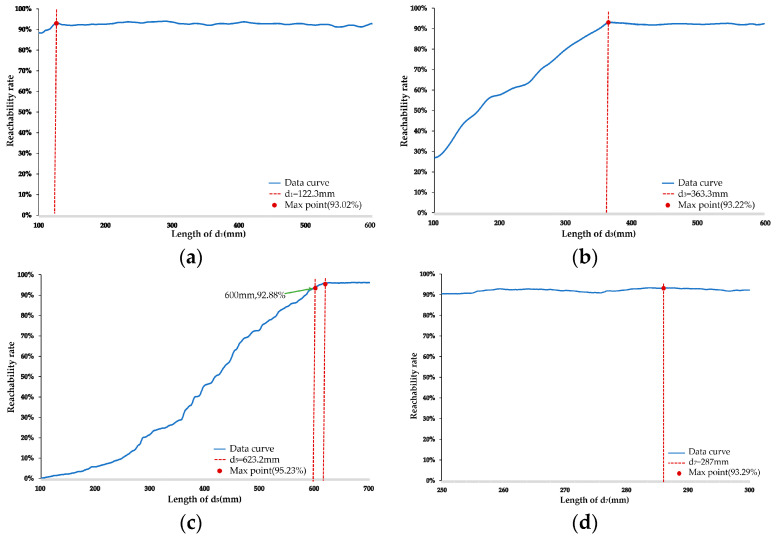
Graph of picking reachability with variation of each arm length parameter. (**a**) Graph of picking reachability with variation *d*_1_ length; (**b**) graph of picking reachability with variation *d*_3_ length; (**c**) graph of picking reachability with variation *d*_7_ length; (**d**) graph of picking reachability with variation *d*_7_ length.

**Figure 8 plants-13-03211-f008:**
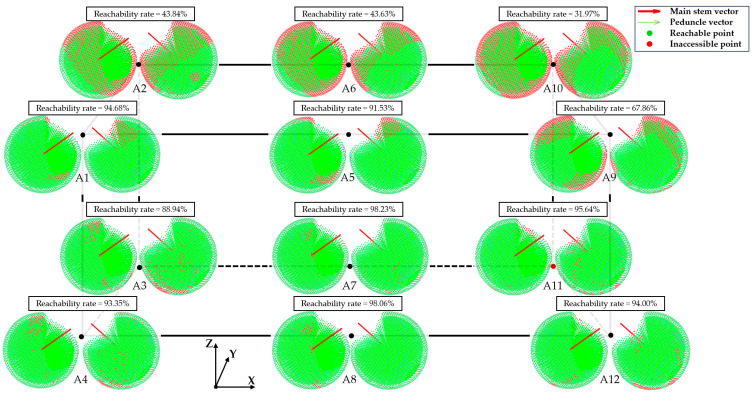
Reachability at spatial points. The left sphere shows a right-front 45° view, and the right sphere shows a left-rear 45° view; the red arrow indicates the main stem vector, the green arrow indicates the peduncle vector, the green point indicates reachable point, and the red point indicates inaccessible point.

**Figure 9 plants-13-03211-f009:**
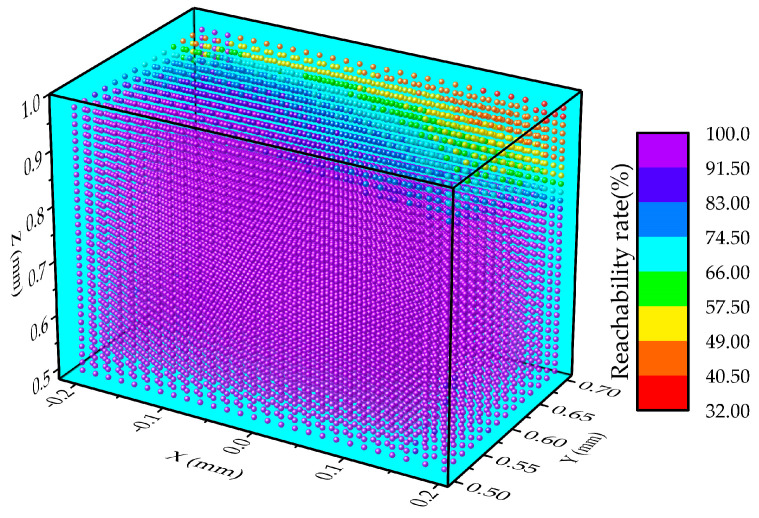
Reachability in the workspace. Red point indicates points with a picking reachability of 32–40.5%, orange point indicates points with a picking reachability of 40.5–49%, yellow point indicates points with a picking reachability of 49–57.5%, green point indicates points with a picking reachability of 57.5–66%, light blue point indicates points with a picking reachability of 66–74.5%, blue point indicates points with a picking reachability of 74.5–83%, dark blue point indicates points with a picking reachability of 83–91.5%, and purple point indicates points with a picking reachability of 91.5–100%.

**Table 1 plants-13-03211-t001:** Tomato posture distribution.

Peduncle Orientation	Front (I)	back (II)	Left (III)	Right (IV)
Quantity	45	42	24	267
Proportion	11.9%	11.1%	6.3%	70.6%

**Table 2 plants-13-03211-t002:** D-H parameters.

*i*	α_*i*−1_	a_*i*−1_	d_*i*_	θ_*i*_
1	0	0	*d* _1_	0
2	π/2	0	0	0
3	−π/2	0	*d* _3_	0
4	π/2	0	0	0
5	−π/2	0	*d* _5_	0
6	π/2	0	0	0
7	−π/2	0	*d* _7_	0

**Table 3 plants-13-03211-t003:** Time consumption and average fitness values under different maximum iteration numbers.

Max Iterations	50	100	200	500	1000	5000
Duration/s	0.0177451	0.035522	0.0712329	0.1711181	0.3473645	1.7279665
Norm	0.011814449	0.002152122	8.88414 × 10^−6^	8.88413 × 10^−6^	8.88413 × 10^−6^	8.884129 × 10^−6^

**Table 4 plants-13-03211-t004:** Performance indicators after optimization.

	*d* _1_	*d* _3_	*d* _5_	*d* _7_	Sum	Reachability Rate
Length (mm)	122.3	363.6	600	287	1372.9	92.88%

## Data Availability

The original contributions presented in the study are included in the article; further inquiries can be directed to the corresponding authors.

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
