# Peer review of "Structural Parameter Optimization of a Tomato Robotic Harvesting Arm: Considering Collision-Free Operation Requirements"

_plants, 2024, doi:10.3390/plants13223211_

Round 1
Reviewer 1 Report
Comments and Suggestions for Authors
The lack of a classic academic writing structure makes the text very difficult to read, to know exactly what was done, how it was done and how it can be evaluated as promising or not. There is no material and methods section. A statistical analysis with comparative academic methodology is not presented. BASED ON THE INFORMATION PROVIDED, THE EXPERIMENT CANNOT BE REPRODUCED, therefore, it is a case study, and should be considered as such.

Reviewer 2 Report
Comments and Suggestions for Authors
I have worked quite a bit in this topic, especially in robotics in agriculture for orange harvesting and for the use of electric robots in agricultural treatments of vineyards.
The work presented is well done and describes well the problem of tomato harvesting with the use of serial robotics, clear in the exposition and in the results. The multi-objective genetic algorithm to optimize the length of the arm has been implemented and the results have been with virtual experiments. The efforts of the authors have been in the optimization of the structural parameters of the robotic arms specialized in fruit and vegetable harvesting and obtaining good numerical results. Unfortunately, my experience teaches, that in this field the experimental test is the true correct final result of the project. Therefore I invite the authors to continue the research in the topic and in the next paper with experimental texts.
I suggest adding the following article:
Cammarata, A., Maddio, P.D., Sinatra, R., Belfiore, N.P. Direct Kinetostatic Analysis of a Gripper with Curved Flexures (2022) Micromachines, 13 (12), art. no. 2172, . DOI: 10.3390/mi13122172
Reviewer 3 Report
Comments and Suggestions for Authors
The paper presents an interesting study that shows the protocol authors used to optimize the geometric parameters of a robotic arm in order to make it best suitable for tomato harvesting operations. The first two sections of the document cover very well basic concepts that help readers understand the context and challenges addressed. Sections 3 and 4 covers the details of the kinematic model and optimization method used in the study, but this time they lack information and formalism/rigor to describe mathematical expressions and the steps performed. Those are the weaker sections of the paper. Later, in the results section, the optimization solution is tested, showing several experiments that validate the statement given by the authors of maximizing the reachability of the arm while keeping it compact. However, in multiple parts of the document, including the title, the term "collision-free" is used, which until reaching the results section I thought was the main goal, but certainly, it was not validated by the results presented. Overall, the paper presents an interesting study but requires revisions in order to be considered for future publication. I am including a list of comments and recommendations for the authors to address in order to improve the quality of their work:
Abstract
-The names of parameters d_1,d_3,d_5, and d_7 are too specific to be placed here before being introduced. I recommend removing them or explaining what are they.
1 Introduction
-The full name Denavit–Hartenberg needs to be introduced before using acronyms D-H.
2 The tomato harvesting environment
-There is an error at the beginning of the section where it is supposed to say Fig 1a. The same error repeats in every mention of the figures in the rest of the document. Please check if that error appears in your original submission.
-The term "collision-free" is being used in the title of the paper and in multiple sections of the paper. But it was never defined properly before being used for the first time. It is important that the authors clearly state what kind of collisions, i.e. what obstacles are interacting and in which situations that may happen. This was mentioned briefly at the end of page 5. Most importantly, how is the "collision-free" term related to the reachability and compactness of the robotic arm in tomato harvesting?
-In Figure 2, was this analysis done with tomatoes from the same row? or samples from different rows in the greenhouse? please clarify.
-Just before Eq. 9, it is not really clear what is the difference between the "target" and "desired" posture vectors.
4 Structural Parameter optimization
- This section requires revision to make it easier to understand the optimization problem. It is not clear what are the objective functions or the steps. It is strongly recommended to use algorithmic representations (e.g. pseudo code representation) to describe the steps clearly and use adequate mathematical notation to represent the parameters used in the optimization process.
- In this section authors mentioned that the objective is to "optimize the geometric parameters of the robotic arm to enhance picking reachability while ensuring a more compact mechanical structure". However, in the title and abstract of the paper, authors state that the aim of the optimization is to satisfy "collision-free picking requirements". Then, how is this accomplished by the proposed optimization approach? which restrictions ensure a collision-free posture or the robotic arm? This is really misleading, the results presented validate the reachability of the arm but I don't see any result that validates the obstacle-avoidance picking. Please clarify this
-There are some typo errors in this and other sections. Please double-check.
-Formatting needs to be improved. For instance, every time a list of steps is described, they should be presented as a list environment or as a diagram. Also, sometimes when authors refer to equations, they use capital letters, sometimes not. Sometimes there is no space between the numbers. Please, be consistent and follow a format.
-How alpha is used in (11)? the mathematical notation within the document needs to be revised.
-I don't see a clear dependency of Eq. (12) and angle alpha. Was that a mistake?
-At the beginning of page 12, there is a mention of objective function 1. This is the first time it was mentioned, but it was never described or introduced.
- Parentheses in (14) could be removed
- The steps of the genetic algorithm should be formalized by using an algorithmic representation and mathematical symbols for every important parameter such as optimal individual coefficient, convergence coefficient, population, and ranks.
5 Results
-It would be important to clarify what means a Failure in Figure 8 and 9?
-Regions are mentioned in the text but not illustrated in Figure 8.
Comments on the Quality of English Language
English writing can be improved and there are some typo errors. Please double-check.
Reviewer 4 Report
Comments and Suggestions for Authors
This paper is about the structural optimization of a tomato robotic harvesting arm. It begins with a short introduction considering relevant SoA works in tomato harvesting and other fruits. In the second section, the greenhouse environment and the collision free requirements of the tomatoes is presented. In the same section, the end effector for harvesting bunches of tomatoes is presented. The orientation of the gripper is analyzed in detail. In section three, the kinematic model of a 7 DoF robot is presented, followed by the inverse kinematics solution using the PSO method. By simulated results it was defined that after 200 iterations the result of the inverse kinematics is successful. In section 4, structural parameters optimization of the harvesting arm is presented. The optimization function model is based on two objective functions and 4 constraints. A MOGA is used to determine the best solution. In function 5 the analysis results of the optimization is presented for maximizing picking reachability of the workspace up to 92.88%.
In summary, it is a very interesting paper but the are a lot of issues to be addressed.
1. First, the complete model of the position of the base of the robot (it is shown in figure 5a, that it is in a mobile platform) and the plants of tomatoes is missing. In fact the determination of O(x_o,y_o,z_o) is missing.
2. The determination of of T_p is based on some points on the main stem and peduncle. Is the distances available to all plants? Is it better to determine directions than points?
3. The kinematics of the robot implies that the O coordinate system is in the base of the robot. Some transformations are missing to complete the loop. This issue is connected with the first one.
4. The structural parameter optimization is again constrained by the base of the robot.
5. The d_5 value of the optimization results violates the third constraint.
6. There are a lot of broken references between the figures, and some wrong references of equations (eq (12) does not contain the angle \alpha) making the study of the paper very difficult.
Some minor issues:
1. Highlight the contribution of the paper. Give more details.
2. Polish typos.
Round 2
Reviewer 1 Report
Comments and Suggestions for Authors
The manuscript is now in a suitable condition for publication.
Reviewer 3 Report
Comments and Suggestions for Authors
The authors have addressed all the comments given. It is now easy as a reader to understand all the steps on their methodology.